# Changes in Protein Degradation and Non-Volatile Flavor Substances of Swimming Crab (*Portunus trituberculatus*) during Steaming

**DOI:** 10.3390/foods11213502

**Published:** 2022-11-03

**Authors:** Qin Chen, Yurui Zhang, Lunan Jing, Naiyong Xiao, Xugan Wu, Wenzheng Shi

**Affiliations:** 1College of Food Science and Technology, Shanghai Ocean University, Shanghai 201306, China; 2College of Fisheries and Life Science, Shanghai Ocean University, Shanghai 201306, China; 3National R&D Branch Center for Freshwater Aquatic Products Processing Technology (Shanghai), Shanghai 201306, China

**Keywords:** *Portunus trituberculatus*, protein degradation, non-volatile flavor, free amino acids, flavor nucleotides, steam

## Abstract

To investigate the effect of steaming time (0, 5, 10, 15, 20, and 25 min) on the protein degradation and non-volatile flavor substances of swimming crab (*Portunus trituberculatus*), the moisture content, total nitrogen (TN), non-protein nitrogen (NPN), free amino acids (FAAs), flavor nucleotides, electronic tongue analysis, and sensory evaluation were determined. The results showed that the contents of NPN and total FAAs were the highest after crabs steamed for 10 min. Meanwhile, the AMP (adenosine monophosphate) content reached the maximum value (332.83 mg/100 g) and the taste active value (TAV) reached 6.67, which indicated that AMP contributes the most to the taste of steamed crab at 10 min. The electronic tongue distinguished the taste difference well, and the sensory score was the highest at 15 min. Combined with equivalent umami concentration (EUC) and TAV value, swimming crab (weight = 200 ± 20 g) steamed for 10–15 min tasted best.

## 1. Introduction

Swimming crab (*Portunus trituberculatus*), one of the most critical marine economic crustacean species in the world, is popular for its succulent meat, ample nutrition, and distinctive flavor [1,2], and the marine capture production in 2020 has amounted to 367 thousand tons [3]. Due to the characteristic of easy corruption, swimming crabs are mainly sold fresh or frozen, which leads to low utilization of production and processing. However, hot-processed items have a higher commercial value in global commercial marketplaces, especially when supply is affected by the season [4]. Hence, it is significant to study the processing quality of swimming crab further to supply high-quality products to consumers and establish a higher-quality market.

Thermal processing is defined as heating meat to a temperature high enough to denature proteins [5], which is an effective measurement to enhance food quality. Thermal processing methods, including microwaving, steaming, boiling, and roasting, have been used to process frozen surimi, sturgeon, shrimp, and mangrove crab, which had a significant effect on flavor formation and consumer performances [6,7,8,9]. Generally, the most common hot processing procedures for crabs are steaming and boiling. Shi et al. [10] investigated the effects of steaming versus boiling on the flavor of swimming crab and discovered that steaming with hot water produced crabs with more water-soluble flavor substances. Simultaneously, steaming time also has a significant impact on the flavor formation and consumer choice of aquatic products. If the steaming time is insufficient, the meat can’t be separated from the exoskeleton adequately. If the steaming time is too long, excessive water loss deteriorates texture; the Maillard reaction reduces the nutritional value. Furthermore, because the quantities of ammonia in crustacean muscles are higher than in vertebrate muscles, superheated steam creates unpleasant ammonia compounds [11]. In several studies, the recommended steaming time for sturgeon and black carp was determined [7,12]. Moreover, the characteristic flavor compounds during steaming have been identified, which provide a theoretical basis for increasing the flavor and promoting the processing of aquatic products. However, the flavor of swimming crab was evaluated mainly in terms of different genders, edible parts, habitat environments, and dietary nutrition [13,14,15,16], and there is little data on the effect of steaming time on the flavor formation of swimming crab. Thus, investigating the flavor change of swimming crab depending on steaming time is of great significance.

Flavor includes taste and smell, and taste is one of the most critical characteristics that influence consumer preference. There are two main types of taste compounds in aquatic products: nitrogen-containing compounds (e.g., free amino acids (FAAs), flavor nucleotides, betaine, trimethylamine oxide, etc.) and others without nitrogen [17]. FAAs are the end products of protein degradation, divided into umami, sweet, and bitter amino acids. Flavor nucleotides, including guanosine monophosphate (GMP), inosine monophosphate (IMP), and adenosine monophosphate (AMP); AMP provides umami and sweetness taste profiles and inhibits bitterness; IMP is merely sweet when the concentration is between 0.5–1.0 mg/mL and shows intense umami flavor when the concentration exceeds 1.0 mg/mL [18]. Additionally, FAAs and flavor nucleotides have synergistic effects on umami and sweetness in swimming crabs and are influenced by heat treatment [19]. Therefore, it is necessary to explore the changes in protein degradation and flavor substances of swimming crab during steaming.

To examine how the swimming crab’s flavor changed during steaming, high-performance liquid chromatography (HPLC) and an ultra-high-speed automatic amino acid analyzer were utilized to analyze nucleotides and FAAs, respectively; the electronic tongue was used to describe the taste modifications; taste quality indicators included the equivalent EUC and TAV. Protein degradation was also discussed. Furthermore, sensory evaluators graded swimming crabs steamed at different times. This study creates the ideal conditions for cooking swimming crab, establishes the link between protein degradation and flavor formation, and provides a theoretical basis for promoting industrial production.

## 2. Materials and Methods

### 2.1. Materials

Monopotassium phosphate, dipotassium phosphate, methanol, amino acid standards, and nucleotide standards were chromatographic reagents purchased from Shanghai Ample Scientistic Instrument Co., Ltd., (Shanghai, China). Analytical grades of sodium hydroxide, potassium hydroxide, perchloric acid, trichloroacetic acid, and other reagents were purchased from Sinopharm Chemical Reagent Co., (Shanghai, China).

### 2.2. Preparation of Sample

Swimming crabs (weight = 200 ± 20 g, n = 60) were bought from Luchaogang in November 2021 in Shanghai, China and transported to the laboratory within 2 h on ice. Crabs were washed clean with tap water and separated into 6 groups at random. Subsequently, the samples were steamed for 5, 10, 15, 20, and 25 min, respectively, in an induction cooker (1600 W power). The time started when the water temperature reached 100 °C, and fresh crabs were served as a control group. After cooling at room temperature, the body meat of each group was collected separately and stored in plastic bags at −80 °C for further analysis.

### 2.3. Moisture Content

The moisture content was measured according to AOAC [20]. Briefly, to measure the moisture content, 2.00 g of the sample was dried to a constant weight at 105 °C in the drying oven (Boxun, Shanghai, China).

### 2.4. Total Nitrogen (TN)

TN was measured according to AOAC. The sample of 0.20 g was weighed and digested for two hours with 12 mL concentrated sulfuric acid, 0.40 g copper sulfates, and 3.50 g potassium sulfates (kjeldahl8400, FOSS, Hillerød, Denmark). The absorbing liquid was 1% (*m*/*v*) boric acid solution, and the titrant was 0.1 mol/L hydrochloric acid. Both the 0.1% (*m*/*v*) bromocresol green and 0.1% (*m*/*v*) methyl red served as indicators.

### 2.5. Non-Protein Nitrogen (NPN)

NPN was determined based on the method of Cambero et al. [21]. The sample of 1.00 g was weighed and homogenized for 2 min in 10 mL of 10% (*v*/*v*) trichloroacetic acid. After standing for 2 h at 4 °C and centrifuged (10,000× *g*, 4 °C, 15 min), 2 mL supernatant was taken and adjusted. The following steps were the same as TN.

### 2.6. Protein Degradation Index (PI)

NPN stands for the non-protein nitrogen content and TN for the overall nitrogen concentration [12]. The percentage ratio between NPN and TN was used to determine PI.
(1)$$\mathrm{PI}=\frac{\mathrm{NPN}}{\mathrm{TN}}$$

### 2.7. FAAs

A minor adjustment was made based on Zhuang et al. [22]. Briefly, the sample of 0.50 g was weighed in a 50 mL tube and homogenized for 2 min with 15 mL of precooled trichloroacetic acid solution (5%, *w*/*v*). After ultrasonic extraction for 15 min, the sample was centrifuged at 10,000× *g* for 15 min at 4 °C before being stored at 4 °C for 2 h. The pH of 2.0 was then adjusted in 5 mL of supernatant using sodium hydroxide solution (3 M and 1 M). A 0.22-μm syringe filter was used to filter the sample solution after the supernatant had been diluted to 10 mL with ultrapure water. An ultra-high-speed automatic amino acid analyzer was used to determine and analyze FAAs (LA-8800, Hitachi, Japan). Each sample was examined three times.

### 2.8. Nucleotides

5′-nucleotides were analyzed according to Zhang et al. [23]. The sample weighed 3.00 g, was placed in a 50 mL tube and homogenized for 1 min in a 10 mL cold solution of 10% (*v*/*v*) perchloric acid before being centrifuged at 10,000× *g* for 15 min at 4 °C. The supernatant was separated from the precipitation, and remained precipitation was rinsed with 5 mL 5% (*v*/*v*) perchloric acid solution before being centrifuged in the same manner. The action mentioned above was carried out twice, and the pH of 5.75 was achieved in the combined supernatant by adding cold potassium hydroxide solution (6 M and 1 M). The supernatant was diluted to 50 mL and then filtered with a 0.22-μm syringe filter for the test after being kept at 4 °C. The entire procedure was carried out below 4 °C. Finally, the analysis was performed by HPLC (Waters Co., Milford, MA, USA). Each sample underwent a triple analysis, and the analytical conditions were performed by Tu, Wu, Wang and Shi [17].

### 2.9. TAV and EUC

The ratio of a flavor ingredient’s concentration to its threshold value is known as the TAV. TAV can be used to determine how much a flavor substance contributes to the overall flavor of samples [24]. When the TAV value is less than one, the sense barely affects the taste. While the value is greater than one, the compound has a significant impact on flavor. The greater the TAV is, the more significantly it contributes to the overall flavor of samples.

EUC is the equivalent of monosodium glutamate (MSG), which represents the intensity of umami produced by the synergistic action of umami amino acids (Asp and Glu) and 5′-nucleotides (IMP, GMP, and AMP). The following formula is the equation for EUC [25]:(2)$$\mathrm{EUC}=\Sigma a_{i}b_{i}+1218\times \left(\Sigma a_{i}b_{i}\right)\times \left(\Sigma a_{j}b_{j}\right)$$
where *a_i_* is the content of umami amino acids (Asp and Glu); *b_i_* is the relative umami concentration (RUC) of each umami amino acid (Asp, 0.077; Glu, 1); *a_j_* is the flavor nucleotides content (IMP, AMP and GMP); *b_j_* is the RUC of the taste nucleotide (IMP, 1; GMP, 2.3; AMP, 0.18) and the synergistic constant is 1218 [26].

### 2.10. Electronic Tongue

A sample (5.00 g) was homogenized for 2 min in ultrapure water (100 mL) using a high-speed homogenizer (Shanghai Fokker Equipment Co., Ltd., Shanghai, China). The sample solution was centrifuged at 10,000× *g* for 15 min at 4 °C after being kept in a cooling bath for 30 min. A 30 mL supernatant was prepared for an electronic tongue on-machine test (ISENSO, Super Tongue, New York, USA), and the entire experiment was conducted under 4 °C. A triplicate of each sample was used for analysis.

### 2.11. Sensory Assessment

Eight sensory assessors (four women and four men, aged 22–25) who are not partial to food or allergic graded the fresh and steamed samples according to the five aspects of order, taste, appearance, morphology, and texture. The specific standards referred to the method of Yang et al. [27] and are shown in Table 1. After tasting 1 sample, the sensory evaluator gargled with pure water and moved on to the next. Each item was rated between 1 and 20 points, and the highest sensory evaluation score for each sample was 100 points. The higher the point is, the more acceptable the sample will be. The Sensory Evaluation Laboratory served as the location for the entire procedure.

### 2.12. Statistical Analysis

All tests were performed three times. Three duplicates of each experiment were evaluated. SPSS26.0 (SPSS Inc., Chicago, IL, USA) was used to analyze the data, and the findings are presented as means and standard deviations (SD). The Duncan technique was used, and *p* < 0.05 was used to denote a significant difference in the one-way analysis of variance (ANOVA). The charts were created by Origin2021 (Origin Lab Corporation, Hampton, Northampton, MA, USA).

## 3. Results and Discussion

### 3.1. Moisture Content Analysis

The moisture content of swimming crab steamed at different times is displayed in Table 2. With the increase in steaming time, the moisture content increased and then decreased. The moisture content of the fresh sample was 79.85% and rose a little after steaming for 5 min (80.84%), which may be due to the accumulation of water vapor in the crab’s meat, and the shell prevents the evaporation of moisture at the beginning of steam. Subsequently, the moisture content decreased slightly and was always lower than that of the fresh sample. The result could be attributed to the protein denaturation, especially the severe contraction of myofibrillar protein after heating, which lowered the muscle’s water-holding ability [28]. In the later stage of steaming, the moisture content was almost unchanged, which might be because the external water vapor content was close to saturation, so the sample water loss was not apparent. Moreover, crab meat contains collagen, which can be turned into gelatin after high-temperature treatment and absorb water and reduce water loss [28].

### 3.2. Protein Degradation Analysis

The changes in TN, NPN, and PI reflect the protein degradation during steaming. The contents of TN and NPN on a dry basis of swimming crab during steaming are shown in Table 2. The TN content increased and then decreased during the steaming process, which reached the highest value (90.12 ± 0.71 g/100 g) after steamed for 5 min and fell as the steaming process continued, which may have resulted from the loss of volatile nitrogen or the dissolution of some of the water-soluble nitrogen into the exudate. The NPN content increased and then decreased during steaming, which was always higher than that of the fresh sample. The result indicated that steaming treatment increased the NPN content of swimming crabs, and the highest value was obtained at 10 min (12.23 ± 1.05 g/100 g). It was mainly due to the protein degradation in the steaming process, which formed a variety of small molecules of peptides, amino acids, and other non-protein substances [29]. These protein degradation products have a significant effect on flavor formation. After continuing to steam, NPN began to decrease. The concentration of NPN in the steamed sample was decreased mainly as a result of the aggregation of FAAs and polypeptides, among which water-soluble amino acids and peptides were transported to the exudate (Zhao et al., 2008).

### 3.3. FAAs Analysis

Aquatic products may have various taste perceptions depending on the components, concentration, and threshold of FAAs [30]. Due to the variations in moisture content during the steaming process, the composition proportion of crab meat was affected. To investigate the specific changes of flavor substances in crab meat with different steaming times, the content of a dry basis is more appropriate. A total of 17 FAAs, including umami amino acids (Glu and Asp), sweet amino acids (Gly, Ala, Thr, Ser, Pro and Arg), and bitter amino acids (Val, Lys, Leu, Ile, Phe, His, Tyr and Met), were displayed in Table 3. During steaming, the amounts of total FAAs increased and then declined, and the content was the maximum (10,996.21 ± 1167.47 mg/100 g) at 10 min, which was consistent with the highest NPN content in crabs steamed for 10 min. In both fresh and steamed samples, the concentration of sweet amino acids was the highest, followed by bitter and umami amino acids. The variance trend for the three different types of amino acids was essentially the same, rising at the start of steaming and then falling. The maximum concentration of sweet amino acids (9418.84 ± 1326.57 mg/100 g) and umami amino acids (520.24 ± 21.34 mg/100 g) were obtained at 10 and 15 min, respectively. In addition, there was no significant change in bitter amino acid content in fresh and steamed samples. This means that steaming doesn’t change the bitter taste of the crab meat; however, the proper steaming time can increase the umami and sweetness. Thr, Gly, Ala, and Pro were the most abundant sweet amino acids in fresh crab, among which the content of Gly (2530.88 ± 89.82 mg/100 g) was the highest. The content of Gly reached the maximum (2622.46 ± 390.54 mg/100 g) after crabs were steamed for 10 min and then decreased after continuous steaming. The changes in Thr, Ala, Pro, and Arg contents were about the same as Gly during steaming. It was reported that the taste of sweetness was noted in Arg at low concentrations (less than twice the threshold) and the taste of bitterness at high doses. However, unlike those hydrophobic amino acids with branched chains that create disagreeable bitterness, Arg can increase flavor complexity and freshness [31]. Asp and Glu were the primary umami amino acids, and the composition of the fresh sample and the sample that had been steamed for five mins did not differ significantly. However, the contents of Asp and Glu reached the maximum at 10 min (144.37 ± 5.17 mg/100 g) and 15 min (409.59 ± 16.41 mg/100 g), respectively. After continuing to steam, the contents of Asp and Glu started to decline. Most bitter amino acids reached their maximum contents when crabs were steamed for 10 min; however, due to the high threshold, bitter amino acids contributed little to the taste quality of fresh or steamed crab. Additionally, several bitter amino acids help to improve sweetness at levels below the threshold [32]. According to Table 3, the primary flavor amino acids of steamed swimming crab are Glu, Gly, Ala, and Arg, and the maximum contents are achieved at 10 or 15 min. After continuous steaming, the content of FAAs began to decrease, which was in line with the trend of NPN during steaming. The increase of free amino acid content in the steaming process was mainly due to protein degradation, however, after continuous steaming, intense aggregation occurred in protein, and the amount of decomposition of Maillard reaction was larger than the amount of thermal degradation of protein, which led to a decrease in free amino acid content in the later period of steaming [33]. In the process of steaming, muscle fibers will contract after being heated, along with a loss of juice. The proteins in the lost fluid are mainly water-soluble proteins and the degradation products of some polypeptide compounds with small molecular weight or myofibrillar proteins. During the heating process, various covalent bonds and non-covalent bonds that maintain protein molecular structure in muscle tissue gradually break, and myofibrilla proteins lose their advanced structure and begin to dissolve. All of these will contribute to the loss of free amino acids in muscle during steaming. In addition, the Maillard reaction, as the main pathway for volatile flavor compound generation, also accelerates FAAs consumption in this process [34]. Therefore, the free amino acid content of swimming crab could be affected by steaming time, and the crab steamed for 10–15 min may have better taste quality.

### 3.4. Nucleotide Analysis

Flavor nucleotides are essential compounds of aquatic products, especially GMP, IMP, and AMP, which provide umami and sweetness taste profiles and synergy with free amino acids to improve the umami taste [18,35]. The dry base content of umami nucleotides (GMP, IMP, and AMP) is shown in Table 4. In the fresh sample, the content of IMP (307.49 ± 18.56 mg/100 g) was higher than the contents of AMP (119.85 ± 22.48 mg/100 g) and GMP (6.7 ± 0.15 mg/100 g). Kong et al. [36] made a similar observation in the Chinese mitten crab. However, the contents of GMP and IMP continued to decrease during steaming, which is contrary to the trend of AMP. The same result was found in Pacific white shrimp and Antarctic krill [37]. Mendes et al. [38] also reported that IMP was mainly found in fish and AMP in crustaceans. Moreover, the GMP content of steamed swimming crab meat decreased sharply and wasn’t detected after 5 min. This result was consistent with the findings of Gorbatov and Lyaskovskaya [39] and Zhang, Qiu, Zhang, Ho Row, Cheng, and Jin [37], who observed that different nucleotides have varying degrees of heat stability, leading to different degrees of thermal degradation. The thresholds of GMP, IMP, and AMP are 12.5 mg/100 g, 25 mg/100 g, and 50 mg/100 g, respectively. In the steaming process, the TAV value of IMP decreased gradually and was always less than one after 5 min. On the contrary, the TAV value of AMP was always greater than one during steaming and also greater than that of the fresh sample, which indicated that AMP contributed much to the flavor of steamed swimming crab, especially the TAV of AMP reached the maximum (6.67) at 10 min. During the steaming process, the total amount of IMP and AMP didn’t change much; however, it decreased obviously at 25 min, which indicated that long-time steaming wasn’t conducive to the formation of flavor. Therefore, swimming crabs should be steamed at a reasonable time to ensure delicious taste.

### 3.5. EUC and TAV Analysis

The EUC value has been universally acknowledged as the primary taste evaluation measure for assessing the umami level of aquatic items [40]. It reveals the direct synergistic action of flavor nucleotides and FAAs of umami and sweet taste and evaluates the flavor of umami enhancement in aquatic products [41]. Figure 1 shows the EUC and TAV values of swimming crab meat throughout steaming. The EUC and TAV values of the fresh sample were significantly higher than steamed ones (*p* < 0.05), and the EUC value was the highest at 4.65 g MSG/100 g in fresh meat. In other words, 100 g of fresh swimming crab meat equals 4.65 g of MSG. During the steaming process, the EUC value declined, and there was no significant change during 5 to 20 min, while it decreased to the lowest (0.16 g MSG/100 g) at 25 min (*p* < 0.05), which meant that steaming for too long had a harmful effect on the formation of flavor. Throughout the procedure, all TAVs of EUC values were significantly greater than one, which indicated that the interaction of nucleotides and FAAs had a significant role in umami taste qualities of swimming crab. This finding is in line with the work of Feng, et al. [42].

### 3.6. Electronic Tongue Analysis

The electronic tongue (e-tongue) is a modern qualitative and quantitative analysis instrument, primarily made up of an interactive sensitive sensor array, a signal acquisition circuit, and a pattern recognition-based data processing approach [43]. E-tongue is now widely used to identify the taste profiles of foods such as livestock and poultry meat, marine products, fruit and vegetable products, etc. Figure 2 shows the PCA chart for the taste profiles of swimming crab meat during different steaming times. According to Garcia Hernandez et al. [44], the two primary components have enough information to accurately represent the sample’s overall characteristics if their cumulative contribution rate exceeds 85%. The first principal component (PC1) and the second principal component (PC2) accounted for 90.1785% and 8.3820%, respectively; the total proportion was 98.56% (>85%), which indicates that each sample’s flavor change information could be accurately recorded. According to Figure 2, there was no crossover among the six samples distributed in the region, which means that the taste of each sample significantly differs from the others, and the e-tongue can distinguish the tastes of swimming crab meat at different steaming times.

### 3.7. Sensory Evaluation Analysis

Figure 3 shows the sensory score of swimming crabs depending on steaming time. The sensory score of steamed samples increased initially and then decreased, but it was always higher than the fresh sample (62.5 ± 3.55). The maximum sensory score (88.9 ± 7.6) was obtained at 15 min and then reduced to 78.8 ± 9.7 at 25 min. From the results, we held that steamed crab is more acceptable than the fresh sample, while over time, steaming reduces the sensory properties. The reason may be that in the steaming process, the protein was degraded continuously, resulting in the accumulation of free amino acids and nucleotides. Some of these umami and sweet amino acids, such as Asp, Glu, Thr, Gly, Arg, and Pro, peaked at 10 or 15 min. Thus, the sensory properties of sweet and umami tastes were significantly influenced. As the steaming continued, the quality suffered from decreased water-holding capacity, cook loss, and muscle shrinkage, which resulted in a decrease in overall acceptance [45]. The longer the steaming time is, the lesser the acceptance will be.

## 4. Conclusions

The changes in protein degradation and non-volatile flavor compounds in steamed swimming crab muscle were analyzed. Generally, throughout the steaming process (0–25 min), the protein continuously degraded to many flavor precursor substances resulting in changes in taste properties. The levels of NPN and total FAAs peaked at 10 min; the amount of sweet and umami amino acids peaked at 10 min and 15 min, respectively. AMP contributed significantly to the taste of steamed crab, and the content of AMP reached the highest at 10 min. Additionally, the flavor profile of the swimming crab was distinguished through the PCA plot of the e-tongue, and the sensory score reached the highest at 15 min. As the steaming time extends, the taste quality deteriorates. In conclusion, 10–15 min is an ideal steaming time for swimming crab (weight = 200 ± 20 g). The findings provide a theoretical foundation for the taste changes in swimming crab during steaming and serve as a reference for further process of swimming crab. However, the dynamic changes of volatile flavor compounds in the steaming process need further study.

## Figures and Tables

**Figure 1 foods-11-03502-f001:**
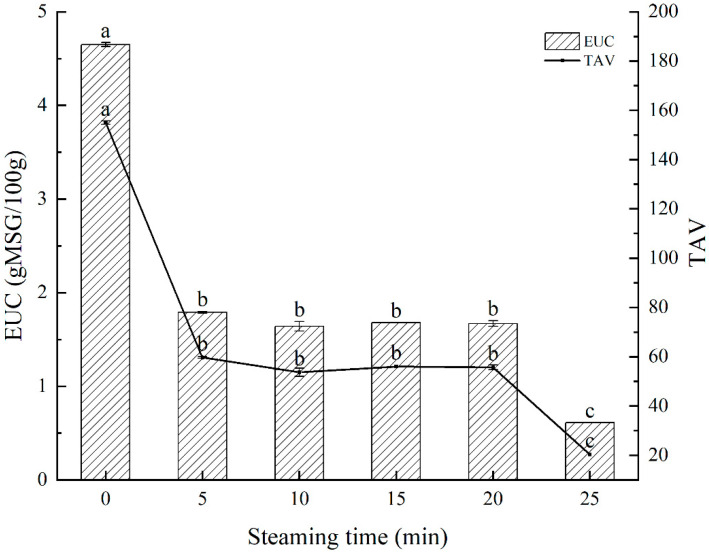
Comparison of EUC and TAV in swimming crab meat during steaming. Different letters are significantly different at *p* < 0.05.

**Figure 2 foods-11-03502-f002:**
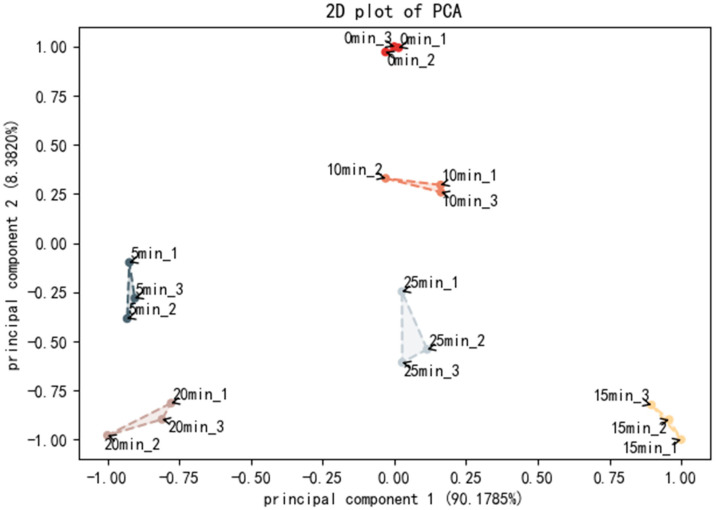
The PCA plot of electronic tongue data of swimming crab steamed at different times.

**Figure 3 foods-11-03502-f003:**
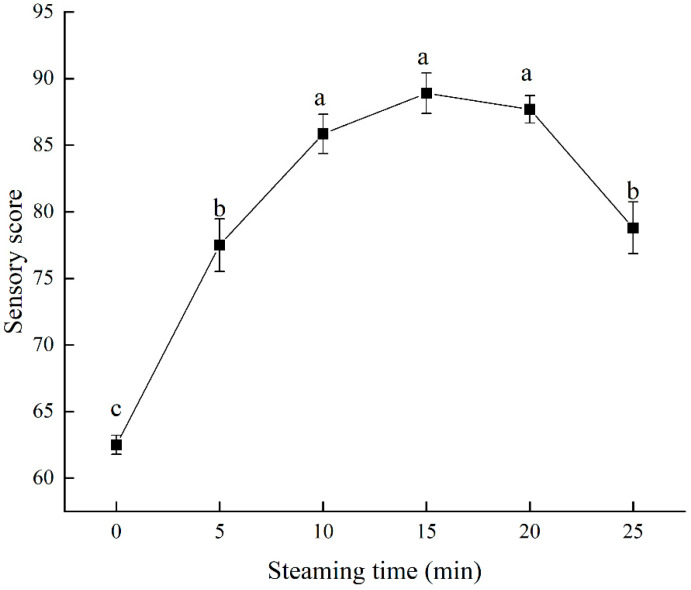
Sensory evolution of swimming crab meat during steaming. Different letters are significantly different at *p* < 0.05.

**Table 1 foods-11-03502-t001:** Sensory evaluation criteria of swimming crab during steaming.

	Score				
Index	1–4	5–8	9–12	13–16	17–20
Order	Unacceptable	No inherent aroma	Slight inherent aroma	Inherent aroma	Strong inherent aroma
Taste	Unacceptable	No inherent umami	Slightly inherent umami	Inherent umami	Strong inherent umami
Appearance	Dim	Slightly dim	Fairly bright	Bright	Translucent
Morphology	Very loose	Loose	Partial loose	Tight	Very tight
Texture	Very soft or stiff	Soft	Slight elastic	Elastic	Firm

**Table 2 foods-11-03502-t002:** Changes of moisture, total nitrogen, and non-protein nitrogen content in swimming crab meat during steaming.

	0 min	5 min	10 min	15 min	20 min	25 min
Moisture content (%)	79.85 ± 0.35 ^b^	80.84 ± 0.07 ^a^	78.28 ± 0.17 ^d^	78.64 ± 0.19 ^cd^	78.68 ± 0.33 ^cd^	78.82 ± 0.25 ^c^
TN (g/100 g)	86.80 ± 2.22 ^b^	90.12 ± 0.71 ^a^	85.19 ± 0.90 ^b^	85.97 ± 0.33 ^b^	82.55 ± 0.91 ^c^	82.06 ± 0.28 ^c^
NPN (g/100 g)	8.88 ± 0.05 ^c^	10.2 ± 0.51 ^bc^	12.23 ± 1.05 ^a^	11.39 ± 0.46 ^ab^	9.19 ± 0.00 ^c^	9.71 ± 0.28 ^c^
PI (%)	10.23	11.32	14.36	13.25	11.13	11.83

Data in the same row with different letters differ significantly (*p* < 0.05).

**Table 3 foods-11-03502-t003:** The contents of free amino acids on a dry basis of swimming crab during steaming (mg/100 g).

		Content (mg/100 g)					
Amino Acid	Threshold (mg/100 g)	0 min	5 min	10 min	15 min	20 min	25 min
Asp	100	109.93 ± 19.74 ^b^	106.5 ± 18.08 ^b^	144.37 ± 5.17 ^a^	110.64 ± 5.24 ^b^	91.24 ± 8.73 ^bc^	73.57 ± 2.24 ^c^
Thr	260	503.59 ± 44.39 ^a^	297.36 ± 51.86 ^b^	436.84 ± 114.29 ^a^	221.49 ± 33.89 ^b^	187.15 ± 29.86 ^b^	295.65 ± 2.31 ^b^
Ser	150	44.72 ± 3.36 ^b^	54.76 ± 3.85 ^a^	25.87 ± 6.43 ^c^	22.17 ± 2.08 ^cd^	17.8 ± 1.53 ^d^	42.59 ± 4.2 ^b^
Glu	30	261.45 ± 10.64 ^bc^	253.01 ± 30.13 ^bc^	329.41 ± 92.37 ^ab^	409.59 ± 16.41 ^a^	409.02 ± 75.62 ^a^	166.92 ± 4.12 ^c^
Gly	130	2530.88 ± 89.82 ^a^	2422.71 ± 168.64 ^a^	2622.46 ± 390.54 ^a^	1757.4 ± 62.35 ^b^	1652.2 ± 243.43 ^b^	1940.14 ± 51.97 ^b^
Ala	60	683.76 ± 27.2 ^a^	432.38 ± 52.15 ^c^	551.5 ± 73.52 ^b^	432.65 ± 44.95 ^c^	671.45 ± 74.86 ^a^	484.54 ± 12.28 ^bc^
Cys	-	14.88 ± 0.36 ^b^	9.72 ± 1.35 ^bc^	12.6 ± 3.89 ^bc^	14.35 ± 1.92 ^b^	23.64 ± 3.99 ^a^	8.24 ± 0.54 ^c^
Val	40	129.99 ± 5.19 ^ab^	96.62 ± 8.44 ^c^	145.61 ± 27.54 ^a^	113.96 ± 18.68 ^bc^	146.41 ± 10.05 ^a^	133.41 ± 3.59 ^ab^
Met	30	182.7 ± 7.19 ^a^	93.61 ± 11.23 ^c^	127.65 ± 24.12 ^bc^	112.24 ± 27.33 ^bc^	182.42 ± 31.89 ^a^	140.94 ± 5.17 ^ab^
Ile	90	55.59 ± 3.04 ^a^	34.1 ± 5.19 ^b^	69.26 ± 19.43 ^a^	53.22 ± 12.79 ^ab^	58.03 ± 4.42 ^a^	59.49 ± 1.09 ^a^
Leu	190	106.35 ± 5.11 ^a^	94.23 ± 9.84 ^a^	137.14 ± 41.15 ^a^	104.06 ± 25.35 ^a^	123.73 ± 11.99 ^a^	113.89 ± 3.37 ^a^
Tyr	-	115.35 ± 3.27 ^b^	159.66 ± 19.27 ^a^	168.45 ± 37.47 ^a^	135.29 ± 20.61 ^ab^	132.5 ± 16.88 ^ab^	150.46 ± 1.02 ^ab^
Phe	90	168.66 ± 2.8 ^a^	167.34 ± 38.02 ^a^	167.51 ± 26.28 ^a^	151.74 ± 17.33 ^a^	153.21 ± 16.41 ^a^	159.11 ± 0.62 ^a^
Lys	50	183.5 ± 7.65 ^a^	152.91 ± 30.99 ^a^	188.84 ± 30.43 ^a^	181.98 ± 24.39 ^a^	172.03 ± 26.51 ^a^	147.18 ± 2.35 ^a^
His	20	92.01 ± 5.89 ^a^	63.4 ± 7.81 ^b^	86.54 ± 16.03 ^a^	65.12 ± 12.11 ^b^	73.91 ± 9.95 ^ab^	88.34 ± 0.23 ^a^
Arg	50	3327.15 ± 71.78 ^bc^	3482.12 ± 261.82 ^bc^	4336.19 ± 925.95 ^ab^	2772.98 ± 88.97 ^c^	3572.72 ± 779.01 ^bc^	4695.08 ± 217.29 ^a^
Pro	300	1132.15 ± 120.49 ^b^	612.5 ± 123.95 ^c^	1445.99 ± 194.25 ^a^	689.76 ± 205.87 ^c^	1024.22 ± 155.7 ^b^	1431.03 ± 75.79 ^a^
Total		9642.67 ± 349.39 ^ab^	8532.94 ± 653.29 ^bc^	10,996.21 ± 1167.47 ^a^	7348.65 ± 576.23 ^c^	8691.67 ± 1249.04 ^bc^	10130.57 ± 236.49 ^ab^
SFAs		8222.25 ± 312.44 ^abc^	7301.83 ± 606.14 ^bcd^	9418.84 ± 1326.57 ^a^	5896.45 ± 424.85 ^d^	7125.54 ± 1202.59 ^cd^	8889.03 ± 255.27 ^ab^
BFAs		1034.15 ± 39.09 ^a^	861.87 ± 109.28 ^a^	1090.99 ± 220.55 ^a^	917.61 ± 157.33 ^a^	1042.22 ± 11.34 ^a^	992.81 ± 17.44 ^a^
UFAs		371.39 ± 30.38 ^bc^	359.52 ± 46.27 ^c^	473.77 ± 89.07 ^ab^	520.24 ± 21.34 ^a^	500.26 ± 84.32 ^a^	240.49 ± 1.88 ^d^

SFAs: the sweet amino acid content (Thr, Ser, Gly, Ala, Arg, and Pro); BFAs: the bitter amino acid content (Val, Met, Ile, Leu, Phe, Tyr, Lys, and His) and UFAs: the umami amino acid content (Asp and Glu). Each provided number corresponds to the mean and standard deviation. Significant differences (*p* < 0.05) are shown by different letters in the same row.

**Table 4 foods-11-03502-t004:** The contents and TAVs of flavor nucleotides on a dry basis of swimming crab during steaming (mg/100 g).

		Content (mg/100 g)			TAV	
Steaming Time (min)	GMP	IMP	AMP	GMP	IMP	AMP
0	6.7 ± 0.15 ^a^	307.49 ± 18.56 ^a^	119.85 ± 22.48 ^d^	0.54	12.3	2.40
5	0.83 ± 0.05 ^b^	105.78 ± 0.05 ^b^	227.99 ± 0.1 ^c^	0.07	4.23	4.56
10	-	18.51 ± 2.76 ^c^	333.54 ± 9.47 ^a^	-	0.74	6.67
15	-	15.26 ± 0.51 ^c^	296.26 ± 2.06 ^b^	-	0.61	5.93
20	-	8.58 ± 0.38 ^e^	332.83 ± 18.88 ^a^	-	0.34	6.66
25	-	9.82 ± 0.05 ^d^	280.78 ± 0.19 ^b^	-	0.39	5.62

Significant differences (*p* < 0.05) are shown by different letters in the same line.

## Data Availability

Data is contained within the article.

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
