# Peer review of "Changes in Protein Degradation and Non-Volatile Flavor Substances of Swimming Crab (Portunus trituberculatus) during Steaming"

_foods, 2022, doi:10.3390/foods11213502_

Round 1
Reviewer 1 Report
It is an interesting and well-planned study. It is planned on a true purpose. It is a comprehensive work with a lot of effort. This manuscript is suitable for publication, as the manuscript is scientifically sound.
My suggestion that it is may have created the thought that I did not read the article thoroughly. However, I can honestly say that I have read the article from beginning to end several times. First of all, I would like to make this point clear. As for the article, it is aimed to investigate protein degradation and non-volatile flavor components during steaming of crab. Crabs are sensitive foods and cooking them is an important issue. There are various studies on the cooking of crabs by various methods. However, I could not find any study on protein degradation and changes in flavor components during steaming. Although there are studies conducted on other aquatic products, I have not seen a study conducted on crab. Therefore, I evaluated the work as original. I also saw that it is a more comprehensive study than other studies. It is a study that deals with the subject in a more comprehensive way. The article is well written and presented throughout. The spelling is pretty good except for a few minor grammatical errors. The article is very easy to read and does not tire the reader. The study results are well presented and interpreted. The discussion is realistic. The results are presented in an understandable way with tables and graphs. Finally, I would like to express that I searched to find a point to criticize from beginning to end and could not find it. I've reviewed it several times to ask for a critique or a correction. I want you to believe in my sincerity. It is a study that includes results that will lead to the cooking of a sensitive product such as crab in the right time without losing its aroma. I think that these results will be useful for crab processors and will contribute to the literature in this sense.
Author Response
you for your positive comments here especially the value of this work. We have carefully revised the manuscript according to the comments. All the changes in the revised manuscript were marked in red.

Reviewer 2 Report
In this study the authors investigated to examine how the swimming crab's flavor changed during steaming by establishing the link between protein degradation and flavor formation and provides a theoretical basis for pormoting industrial production.
As overall comments I would like to say that the data and information are greatly useful for further investigation of application of by product for further use. However, there is a lot of rooms for improvement, as well as, better discussed in some issues. Some issues should be addressed before publication. I suggest some minor revision for publication, including below points:
<Materials and methods>
line 95 : Please add detail temperature of 'living state' for better understanding
line 95 : Please add what type of water did you used for washing tap?? destilled??
line 97 : What the temperature of induction cooker during steaming??
line 104 : Please add the information (model, city, country, etc) of drying oven for moisture content
line 138 : Please change '(6M, 1M)' to '(6M and 1M)'
line 157 : Please change '(IMP, AMP, GMP)' to '(IMP, AMP, and GMP)'
line 161 : Please add the information (model, city, country, etc) of homogenizer
line 163 : Please add city name of ISENSO for electronic tongue machine
line 165 : Actually the number of assessors was too small. Are they well trained assessors?? If it was please describe it
line 184 : Was the moisture content of samples measured after steaming??? not during steaming. If it was right please revise it through the whole manuscript (Table and Figures too)
line 222-223 : Please add 'and' before 'ASP', 'Arg', and 'Met', respectively
line 261 (Table 2) : Please double-check the result of statistical analysis ('a', 'b', 'c', etc)
line 264 : Please add ',' before 'and His)'
line 272 : Please change 'than AMP' to ' than the contents of AMP'
line 343 : Please change 'during steaming' to 'depending on steaming times'
Please double-check the 'References' following the guideline of the Journal
Author Response
Response:We thank the reviewer for the positive and summative comments here especially the value of this work. The manuscript has been carefully modified based on your comments. All the changes in the revised manuscript were marked in red.
Please see the attachment.

Reviewer 3 Report
Almost all proteins are denatured when heated to a temperature of 50 C to 55 C or higher. The mechanism of thermal denaturation is related to the restructuring of the protein molecule, as a result of which the protein loses its native properties and solubility. From this there may be losses of amino acids. In addition, when heated, the amino acids and reducing sugars generate melanoidins, which are involved in the formation of taste, odor, and color in food products.
When describing changes in the amino acid composition of meat, these points should be emphasized.
Author Response
Response: Thank you for your suggestion. We have modified the related contents.
Please see the attachment.
